# Intramuscular Botulinum Toxin for Complex Regional Pain Syndrome: A Narrative Review of Published Cases

**DOI:** 10.3390/toxins17070350

**Published:** 2025-07-11

**Authors:** Marc Klee, Nilkolaj la Cour Karottki, Bo Biering-Sørensen

**Affiliations:** 1Movement Disorder and Pain Research Center, Rigshospitalet Glostrup, 2600 Glostrup, Denmark; nikolaj.la.cour.karottki@regionh.dk (N.l.C.K.); bo.biering-soerensen@regionh.dk (B.B.-S.); 2Faculty of Health and Medical Sciences, University of Copenhagen, 1165 Copenhagen, Denmark

**Keywords:** Complex Regional Pain Syndrome, botulinum toxin, narrative review

## Abstract

Background: Since the 1980s, numerous case reports have explored the use of intramuscular botulinum toxin (BoNT) for Complex Regional Pain Syndrome (CRPS), with significant variation in rationale, dosing, guidance techniques, and outcome measures. This narrative review aims to summarize published evidence on the use of intramuscular BoNT in patients with CRPS, including studies using earlier terminology such as reflex sympathetic dystrophy (RSD). Given the heterogeneous and largely anecdotal nature of the literature, this review is intended to map the existing landscape rather than conduct a formal analysis. Methods: The PubMed and EMBASE databases were searched in August 2024 using terms related to CRPS and botulinum toxin. Following abstract and full-text screening, 25 publications were included. Results: The included studies span single case reports, case series, and small cohorts, encompassing at least 96 individual CRPS patients treated with intramuscular BoNT. Reported outcomes were heterogeneous, and key treatment parameters—such as toxin type, target muscles, guidance technique, and dosing—were inconsistently reported. Conclusion: The evidence for intramuscular BoNT in CRPS remains limited and heterogeneous, preventing firm conclusions on its efficacy or safety. Its use may be considered in select cases, particularly those with disabling or painful focal dystonia or myofascial pain, but standardized prospective studies are needed to clarify its clinical role.

## 1. Introduction

Complex Regional Pain Syndrome (CRPS) is a multifaceted pain condition often accompanied by severe motor dysfunction, including dystonia, spasm, and fixed posturing [1]. These motor features can be as disabling as the pain itself, significantly limiting mobility, daily function, and quality of life [2]. Despite their clinical relevance, effective treatments for CRPS-associated muscle overactivity remain limited. Botulinum toxin is widely used in other movement disorders and pain syndromes [3,4] and seems an intriguing therapeutic option due to its combined analgesic and muscle-relaxant effects.

While its use in CRPS has not been formally studied in large trials, five decades of case reports and small series have described the off-label application of intramuscular BoNT in this setting. However, these reports are heterogeneous in methodology and often lack standardization in dosing, technique, and outcome assessment. Given the chronicity, complexity, and treatment resistance seen in many CRPS cases, there is a need to clarify whether BoNT offers meaningful benefit for selected patients—particularly those with focal dystonia or myofascial pain.

The current diagnostic criteria for CRPS were established in 2003 in Budapest; however, the disease has historically been described using other terms, most notably reflex sympathetic dystrophy (RSD). In this review, we include all relevant reports using intramuscular BoNT for CRPS or synonymous diagnoses, regardless of terminology, methodology, or evidence quality, to provide a comprehensive overview of published clinical experience.

This narrative review synthesizes the available case reports and series chronologically, highlighting key trends, methodological limitations, and clinical observations. It is not a systematic review, as the evidence is too sparse and heterogeneous for formal analysis. Instead, our aim is to clarify the current landscape and emphasize the need for more standardized reporting and study design in future research. To our knowledge, no prior reviews have focused specifically on this application.

## 2. Overview of Published Cases on Intramuscular BoNT in CRPS

Published cases of intramuscular BoNT in CRPS are limited and methodology, dosages, guidance techniques, and outcomes reported vary considerably. Here we present a brief overview of publications, discussed in chronological order. A summary of all included studies is presented in Table 1, detailing publication year, study type, patient numbers, age, gender distribution, diagnosis, and disease duration.

The first published instance of a case of CRPS-related dystonia is a 1988 paper by Jankovic and Van der Linden who reported using BoNT injections in the forearm of a patient who experienced discoloration, swelling, severe pain, and dystonic flexion of the hand, following minor trauma [5]. They reported moderate relief of the flexor spasms but no functional improvement. No further data was reported and the diagnosis of CRPS is not confirmable retrospectively.

Dr. Daniel Tarsy and colleagues published a series of three cases of limb dystonia following high-voltage electrical injury [6]. Of these, one patient is described as having symptoms of reflex sympathetic dystrophy (RSD), including persistent hyperextension of digits 2–5 and limited voluntary movement. Electromyography (EMG) confirmed the presence of “spasms” in the lumbricals, which were subsequently treated twice with 60 units (15 each lumbrical) of an unspecified botulinum toxin, spaced two weeks apart. Passive flexion improved, though active function remained limited.

In 2001, van Hilten et al. published a series of 10 patients with progressive dystonia due to reflex sympathetic dystrophy [7]. BoNT treatment is mentioned briefly as being ineffective in all cases, but no further details are provided.

The same year, a case series of 14 patients with dystonic clenched fist was published by Cordavari et al. [8]. Four patients had CRPS (using the 1994 criteria [30]) and were treated using EMG-guided injections of abobotulinumtoxin-A into wrist and finger flexors. Injections were performed every 3–6 months and treatment was repeated up to 14 times, with changes in dosage and injected muscles according to clinical presentation. Three of four patients experienced consistent pain relief and muscle relaxation, while meaningful improvements in posture and function were only seen in the hand of one patient, whose clenched fist manifestation was treated within 12 months of onset. Dosages were relatively high (up to 300 units per muscle and 1200 units total) and effects “short-lasting and less satisfactory than in patients with Parksinson’s disease”.

In 2002, Argoff and colleagues published an open-label study of 11 patients with type 1 CRPS of the upper limb (originally presented at the 1999 IASP Congress [9], later expanded on by the authors in the referenced publication) [10]. Onabotulinumtoxin-A was injected into myofascial trigger points in the proximal neck and shoulder of the affected limb, resulting in substantial pain relief both locally and distally. Additional benefits included improved range of motion, reduced thermal and mechanical allodynia, and decreased hyperalgesia. Doses of up to 300 units, with 25–50 units per injected muscle were used, with effects lasting up to 12 weeks. Consistent efficacy was reported for up to 4 years of continual treatment using this approach. A randomized, placebo-controlled trial was reportedly initiated, but its outcome remains unknown, and follow-up attempts with the author were unsuccessful.

A larger study from 2004 by Schrag et al. evaluated 103 patients with fixed dystonia, 41 of whom were assessed prospectively [11]. Twenty percent of patients met diagnostic criteria for CRPS, and the study focused on the perceived overlap between fixed dystonia, CRPS, and psychogenic or somatoform disorders. BoNT injections were among the treatment modalities attempted, and only eight patients reportedly experienced improvement ranging from transient benefit to near-complete remission. The authors do not report on whether these eight had a diagnosis of CRPS. The paper also provides no specifics regarding the botulinum toxin type, injection sites, dosing, or outcome measures. Furthermore, the classification of CRPS-associated dystonia as potentially psychogenic is controversial, especially given the growing recognition of CRPS as a condition involving both central and peripheral pathophysiological mechanisms. This lack of detailed treatment data and the psychogenic framing of CRPS-related dystonia limits the generalizability and clinical applicability of the findings.

In 2005, a Brazilian group published a report of two cases of long-standing refractory CRPS type 1 of the upper limb, complicated by an inability to open the hand [12]. Following a course of frequent stellate ganglion blocks with lidocaine and clonidine, they injected 75 units of BoNT in the finger and wrist flexors. The authors reported marked decreases in overall pain and improvements in motor function for both patients.

Another 2005 publication by a French group reported briefly on five patients with RSD [13]. Four women and one man (mean age 52) with RSD of the lower extremity developed tonic dystonia characterized by fixed equinovarus deformity of the foot and abnormal great toe posturing. BoNT injections were administered following motor block in three patients. One experienced minimal effect, while two showed slight improvement in muscle contraction. Specific injection sites were not detailed in the report.

In a 2010 case report, Safarpour and Jabbari used a trigger-point approach to treat two patients with refractory type 1 CRPS [14]. Each received 200–240 units of BoNT-A (toxin type not specified) injected into proximal neck and shoulder girdle muscles. Both patients experienced substantial improvements in proximal and distal pain, muscle tension, range of motion, and daily functioning. Allodynic areas, discoloration, and swelling were reduced within a month, with effects maintained through quarterly injections. No adverse events were noted.

A frequently cited work by Kharkar et al. from 2011 presented a retrospective case series of 37 patients with dystonia in the neck or upper limb girdle [15]. All patients suffered from CRPS of at least one upper extremity (the authors reported that 73% of those included had CRPS of the entire body) and reported severe localized pain of the neck or upper back at baseline. Injections were performed using an EMG-guided technique and each muscle was injected with 10–20 units of onabotulinumtoxin-A. Most patients were injected unilaterally, primarily in the superficial neck muscles. Criteria for choosing injection sites, laterality, dosage, and how the presence of dystonia was verified are not reported. Outcome was measured as the difference in localized pain on an 11-point Likert scale at 4 weeks post-treatment. A total of 97% of patients reported some degree of pain relief, with an average reduction of 43%. No data on functional improvement was included. A single adverse event in the form of head-drop was reported, which resolved spontaneously.

Mauruc et al. published a single case report in 2012 that describes a 42-year-old woman who developed upper limb CRPS with associated dystonia three years after a left humerus fracture and ulnar nerve compression [16]. Initial treatment with regional nerve blocks and rehabilitation yielded no improvement. BoNT was subsequently injected into the trapezius, teres major, pectoralis major, pronator teres, flexor carpi radialis, and the superficial and deep flexors of the second to fifth fingers. Alcohol nerve blocks were also administered to the palmar muscles, common flexors, and flexor pollicis longus. This combination led to measurable and significant functional improvement in shoulder, elbow, wrist, and finger mobility.

In 2013, Vogt, Birklein, and Geber published a case series of six patients with movement disorders secondary to CRPS, including one with facial dyskinesia, three with upper limb dystonia, and two with lower limb dystonia [17]. Four patients had CRPS type I. All were treated with repeated intramuscular injections of BoNT-A, with dosage and injection sites and guidance techniques not detailed. Five of six patients experienced reduced muscle tone, pain relief, and decreased need for analgesics, though full restoration of motor function was not achieved. The authors conclude that BoNT-A is a partially effective, non-invasive treatment option that may be considered before more invasive therapies such as intrathecal baclofen.

Published in *Pain Medicine* in 2014, this case report by Vas and Pai describes a 39-year-old woman with CRPS type 2 following humerus surgery [18]. As part of a multimodal regimen, she received 40 units of BoNT into the interossei and extensor digiti minimi for persistent stiffness and clawing. While overall recovery was attributed primarily to dry needling and physical therapy, botulinum toxin was included among the interventions used during symptom management.

Also in 2014, Fallatah reported using intramuscular BoNT-A injections into the trapezius muscle as part of a multimodal approach to CRPS of the upper limb, which also included brachial plexus blocks, intensive physical therapy, and transcutaneous electrical nerve stimulation [19]. The patient made a complete recovery although the authors attribute only the relief of the proximal myofascial pain to BoNT-A treatment.

A poster at the AAPM&R 2016 annual meeting by Drakeley, describes a 31-year-old man with CRPS and dystonic posturing of the left foot following posterior tibial tendon repair [20]. After failing multiple outpatient treatments, the patient was admitted for inpatient rehabilitation. He received EMG and ultrasound-guided onabotulinumtoxin-A injections into the flexor digitorum brevis (70 units), flexor digitorum longus (70 units), and flexor hallucis longus (60 units). These injections led to marked functional improvement, with ambulation increasing from 11 to 200 feet at discharge. The patient returned to work as a physician, though symptoms recurred at three months post-injection, consistent with the expected duration of botulinum toxin’s effect. The authors conclude that their findings support onabotulinumtoxin-A as a potentially effective component of a multimodal treatment strategy for CRPS-associated dystonia.

Also in 2016, Buonocore et al. described a 47-year-old woman who developed CRPS type II with disabling neuropathic pain following an iatrogenic tibial nerve transection during ankle surgery [21]. Despite extensive pharmacologic, interventional, and neuromodulation therapies—including sympathetic blocks and spinal cord stimulation—her symptoms persisted. Seven years post-injury, she developed painful dystonic posturing involving the tibialis posterior and flexor digitorum longus. EMG-guided injections of abobotulinumtoxinA (120 units per muscle) into these and other tarsal tunnel-associated muscles produced significant, though temporary, pain relief, gait improvement, and reduced frequency of dysesthesic sensations. These effects attributed the improvements to a decrease in direct mechanical stimulation of a neuroma via targeted muscle relaxation.

Altonji et al. presented a complex case of a 48-year-old woman with CRPS type I of the upper limb following a wrist sprain, later complicated by secondary dystonia at the 2017 Neuromodulation meeting in Las Vegas [22]. After failed conservative treatment, she received a cervical spinal cord stimulator with good pain relief, followed by focal onabotulinumtoxin-A injections for dystonia with initial success. Eighteen months post-injury, she developed CRPS in the contralateral lower limb with similar progression to secondary dystonia. A second spinal cord stimulator and intrathecal baclofen (ITB) provided broader relief, while BoNT-A continued to be used for residual focal dystonia. The combined approach enabled her to return to full-time work and independent mobility. Injection sites, doses, and guidance techniques were not specified.

Presented at the American Academy of Pain Medicine in 2018, a case report by Reddy and Ahadian describes a 56-year-old woman with a 17-year history of CRPS type I affecting the right lower extremity, complicated by acquired dystonia of the calf muscles and severe functional impairment [23]. Despite prior treatments—including physical therapy, spinal cord stimulation, and high-dose opioids (370 mg/day)—the patient remained highly symptomatic. BoNT-A was injected into the right gastrocnemius and soleus muscles every 3 months (up to 75 units per session). Over three years, she experienced consistent 80% improvements in pain and function, along with reduced opioid use (down to 7.5 mg/day), improved gait, and resolution of cognitive and affective symptoms. The authors conclude that the treatment is a safe, well-tolerated option for CRPS-related dystonia, even in chronic, refractory cases, and advocate for further systematic investigation.

Shenouda and Gayed presented a case at the AAPM&R Annual Meeting in 2020 concerning a 19-year-old athletic woman who developed CRPS type II with severe equinovarus dystonia following arthroscopic knee surgery [24]. Her symptoms included persistent plantarflexion, inversion, and toe flexion of the right foot, unresponsive to serial casting, physical therapy, sympathetic blocks, spinal cord stimulator trial, orthotics, and even tendon lengthening surgery. Electrodiagnostic testing suggested deep peroneal nerve injury. She experienced transient improvement with tibial nerve blocks under ultrasound and electrical stimulation guidance. Botulinum toxin injections were planned as part of further treatment, but the report does not specify the type of toxin, dose, target muscles, or anticipated outcome measures, limiting its utility in guiding similar interventions.

Published in *The Bone & Joint Journal* in 2021, this retrospective case series by Gray et al. reviewed 29 patients diagnosed with functional dystonia of the foot and ankle, of whom 9 also met diagnostic criteria for CRPS type I [25]. While the primary focus of the study was on functional dystonia of the foot and ankle, it is noted that 20 patients—including an unknown number of those with CRPS—were treated with botulinum toxin injections. Only one patient of twenty reported improvement, and even then, only in conjunction with casting and physiotherapy. Like the 2004 publication on the same topic, this report provides no details on the dosing, injection sites, guidance techniques, or standardized outcome measures, limiting the interpretability and reproducibility of its findings regarding BoNT efficacy.

Chokshi et al. presented a case in 2022 describing a 49-year-old woman with post-operative knee contracture and CRPS following meniscectomy [26]. Post-operatively the patient developed problematic decreased range of motion and pain with active and passive knee flexion, despite full muscle power. BoNT-A was injected into the quadriceps and gastrocnemius muscles, resulting in reported improvements in pain and ambulation.

Although not the focus of the publication, a surgical case report from 2022 by Kuah and colleagues noted a lack of response to botulinum toxin injections in the case of a 39-year-old woman with dystonic rigid adductovarus rearfoot secondary to CRPS of the lower extremity, but provided no further treatment details [27].

Tombak and colleagues published a report of a 53-year old woman who developed CRPS type I with fixed dystonia of the left arm and hand, following a distal radius fracture and lengthy immobilization [28]. The authors describe a pronated, flexed elbow and wrist and flexion of the metacarpophalangeal (MCP) and proximal interphalangeal (PIP) joints. A combined regimen of analgesic and supportive medications, intensive physical therapy including transcutaneous electrical stimulation, and repeated stellate ganglion blocks were able to improve range of motion and pain in the arm and wrist but had no effect on the MCP or PIP flexion. Using 50 units of an unspecified BoNT-A diluted to 10 units/mL, they injected three sites in the thenar region, two in the hypothenar region, and five in the intraphalengeal spaces at the level of the MCP. No further details on guidance, depth, or distribution of the dosage are included. Three weeks post-treatment, substantial improvements in finger range of motion, grip strength, and overall pain were noted, lasting six months.

The most recent publication is a case series on peripherally induced movement disorders in 2024 by Peresa et al. [29]. The report describes a 25-year-old woman (Case 2) who developed CRPS with fixed dystonia following a lateral ankle sprain and arthroscopic anchor repair for an anterior talofibular ligament tear [29]. Two months after surgery, she developed dystonia of the foot with a striatal toe, which gradually spread proximally, resulting in ankle dorsiflexion, inversion, and stiff-knee gait. BoNT-A was administered under ultrasound guidance to the rectus femoris, gracilis, tibialis posterior, tibialis anterior, and extensor hallucis longus muscles. While the primary treatment goal—reduction in striatal toe discomfort—was fully achieved (Goal Attainment Scale (GAS) 0), secondary goals of improving foot contact during stance and reducing stiff-knee gait showed only partial improvement (GAS −1). The report provides moderate detail on muscle targets and goal setting but lacks information on BoNT-A dosage and long-term functional outcomes.

## 3. Discussion

The current literature on intramuscular BoNT for CRPS reveals significant methodological inconsistencies that complicate the interpretation of its efficacy and safety. Published clinical reports vary considerably in terms of their rationale for using BoNT, the specific indications targeted, and the methodological quality of their interventions. Some describe focal injections for dystonia, others target proximal myofascial pain syndromes or fixed postures. Muscles targeted, injection techniques, and doses used are inconsistently reported, with many studies omitting key details such as the type of botulinum toxin, the use of ultrasound or EMG guidance, and the criteria for dose selection. Dosage per muscle and cumulative treatment dose often vary widely, and in several cases, this information is not reported at all—making it difficult to interpret safety or efficacy (Appendix A, Table A1). For other indications, such as cervical dystonia, injection technique has been shown to significantly affect both efficacy and adverse event profiles [31]. This lack of detail hinders both clinical translation and the development of best-practice recommendations.

Equally problematic is the lack of standardized outcome measures. Few studies utilize validated instruments such as pain scales, functional assessments, or range of motion metrics. Where outcomes are reported, they are frequently limited to qualitative descriptors or subjective patient impressions. Follow-up periods are also inconsistent; while some studies report outcomes within days or weeks, longer-term follow-up beyond three months is rare, although several studies reported continuous successful treatment for years (Table 2).

Adverse events, or the lack thereof, was only reported in three publications [14,15,23], and none reported on post-injection symptom flare-ups or exacerbations—an important omission given the heightened sensitivity of CRPS patients to invasive procedures. This lack of adverse event reporting limits our understanding of the risk profile associated with intramuscular BoNT in this population and how to ameliorate it.

Nonetheless, a subset of reports does describe clinically meaningful reductions in pain and improvements in function. These outcomes appear most consistent in settings where BoNT is delivered as part of a multimodal strategy, often in conjunction with physiotherapy, pharmacological management, and regional blocks [19,20,22]. Conversely, isolated use of BoNT—particularly in cases with long-standing fixed deformities—appears less effective [7], although meaningful benefit has been described even after many years of symptomatic dystonia [23]. As demonstrated by Argoff [10], Safarpour [14], and others, treatment of proximal myofascial pain may confer more than just local pain relief, although whether the observed improvements were directly related to the treatment remains unclear. Though not formally part of diagnostic criteria, proximal myofascial pain may affect up to 80% of CRPS patients [32,33,34] and treating it directly could be an important part of a multimodal approach [1,35]. Although mechanistic insights were beyond the scope of this review, some authors have hypothesized that BoNT may exert effects beyond local muscle relaxation, potentially including modulation of nociceptive processing or central sensitization.

The timing of intervention may also influence outcomes. In the Cordivari series, the only patient with sustained functional improvement received treatment within 12 months of symptom onset [8]. Several other authors have similarly noted improved responses when BoNT is administered before postural abnormalities become structurally fixed, a phenomenon well-known from spasticity treatment.

In response to a pervasive perception of BoNT-A’s ineffectiveness in CRPS-related dystonia, Schilder and colleagues performed an electrophysiological study in 2014 to assess neuromuscular responsiveness [36]. Their findings of near-normal pharmacodynamic response in CRPS patients with tonic dystonia contradict the assumption that poor clinical outcomes are due to altered receptor physiology, instead pointing toward alternative explanations such as inadequate dosing, suboptimal injection targeting, or patient selection.

An additional consideration is the classification of CRPS-related dystonia as functional. As noted by Schrag et al. [11] and Gray et al. [25], some authors have interpreted the fixed postures and movement limitations observed in CRPS patients as psychogenic in origin. While there is possibly some degree of phenotypic overlap with functional dystonia, CRPS is now widely recognized as a condition involving both central and peripheral mechanisms, including inflammation, sympathetic dysfunction, and maladaptive neuroplasticity [1]. The presence of dystonia and/or tremor is even included in the Budapest criteria used to diagnoses the condition. It is likely that the dystonia observed in CRPS represents a distinct phenomenon and the psychogenic framing risks under-treatment and diagnostic overshadowing.

Previous reviews on BoNT for CRPS generally support its potential benefit while consistently emphasizing the lack of high-quality evidence [10,15,37]. The most recent, published in 2022 [37], included all clinical trials regardless of injection methodology. Of the eight studies reviewed, only three were suitable for quantitative analysis. As the authors noted, the current evidence base lacks robust, systematic trials. In the absence of high-level evidence, case-based literature becomes a natural point of reference. This narrative review contributes by synthesizing that body of work, aiming to help clinicians make more informed decisions in the context of real-world treatment variability.

## 4. Conclusions

This review presents a chronological overview of published cases involving intramuscular BoNT in CRPS. Overall, the currently available evidence is insufficient to draw any meaningful conclusions on the efficacy and safety of intramuscular BoNT. While it undoubtedly sees regular use in clinical practice, including in our own clinic, the heterogenous nature of the published cases highlights substantial differences in clinical practice across different centers and makes any evidence-based general recommendations difficult. However, attempting BoNT may be relevant in patients with dystonia contributing to functional impairment or pain, or in managing refractory myofascial pain.

Prospective studies using standardized outcome measures, consistent guidance techniques and muscle selection, stratification by chronicity, and protocols incorporating intramuscular BoNT as part of a broader multidisciplinary strategy are needed to clarify its place in the therapeutic arsenal.

## 5. Limitations

This review is limited by the quality, consistency, and volume of the available literature. Most included reports are single cases or small series and in many of them, essential details are not reported. Diagnostic criteria for CRPS were not consistently applied across reports, with several cases using outdated terminology (e.g., reflex sympathetic dystrophy, algodystrophy) or failing to specify diagnostic methods entirely. This variability limits the generalizability of findings and complicates efforts to evaluate safety or efficacy systematically. Publication bias toward positive outcomes must also be considered. Other routes of administration of BoNT, such as intra-articular [38], subcutaneous [39], perivascular [40], and as sympathetic blocks [41], are not within the scope of this review, although a similar number of published cases are available.

## 6. Materials and Methods

The PubMed and EMBASE databases were searched on 13–14 August 2024 using the following terms:


*(“Complex Regional Pain Syndromes” [Mesh] OR “Complex Regional Pain Syndrome” OR “CRPS” OR “Reflex Sympathetic Dystrophy” OR “Reflex Sympathetic Dystrophy Syndrome” OR “Causalgia”)*



*AND*



*(“Botulinum Toxins” [Mesh] OR “Botulinum Toxins, Type A” OR “botulinum toxin” OR “botox” OR “botulinum neurotoxin”)*


This yielded a total of 263 publications, of which 41 passed abstract screening by two reviewers. Only studies which explicitly included intramuscular BoNT treatment were included. Cross-referencing provided a further two publications fit for inclusion. After full-text screening, a total of 25 publications were extracted. Of the 24 publications included, 15 were single case reports, 8 were case series, and 3 were retrospective or prospective cohorts. A total of at least 96 individual CRPS patients received intramuscular BoNT treatment.

## Figures and Tables

**Table 1 toxins-17-00350-t001:** Characteristics of included studies and patients.

Study (Author, Year)	Study Design (Number of Patients)	Age (Range)	Gender Distribution M/F	Diagnosis	Disease Duration (Range)
Jankovic [5] (1988)	Case report (n = 1)	33	0/1	RSD	2 years
Tarsy [6] (1994)	Case report (n = 1)	29	1/0	RSD	N/R
van Hilten [7] (2001)	Case series (n = 10)	32 (18–50)	N/R	RSD	2–24 years
Cordivari [8] (2001)	Case series (n = 4)	36–56	0/4	CRPS	18 months–10 years
Argoff [9,10] (1999, 2002)	Prospective, open-label (n = 13)	N/R	N/R	CRPS I	N/R
Schrag [11] (2004)	Retrospective/prospective (n = 8, unverified)	N/R	N/R	CRPS	N/R
Lauretti [12] (2005)	Case series (n = 2)	38–42	0/2	CRPS I	3.5 years
Morelet [13] (2005)	Case series (n = 3)	34–75	1/3	RSD	N/R
Safarpour [14] (2010)	Case series (n = 2)	41–48	0/2	CRPS I	2–4 years
Kharkar [15] (2011)	Retrospective study (n = 37)	N/R	2/35	26 CRPS I 11 CRPS II	N/R
Mauruc [16]. (2012)	Case report (n = 1)	42	0/1	N/R	3 years
Vogt [17] (2013)	Case series (n = 6)	N/R	N/R	4 CRPS I	N/R
Vas [18] (2014)	Case report (n = 1)	39	0/1	CRPS I/II	3 months
Fallatah [19] (2014)	Case report (n = 1)	34	0/1	CRPS I	5 months
Drakeley [20] (2016)	Case report (n = 1)	32	1/0	N/R	N/R
Buonocore [21] (2016)	Case report (n = 1)	47	0/1	CRPS II	18 months
Altonji [22] (2017)	Case report (n = 1)	48	0/1	CRPS I	N/R
Reddy [23] (2018)	Case report (n = 1)	56	0/1	CRPS I	17 years
Shenouda [24] (2020)	Case report (n = 1)	19	0/1	CRPS II	N/R
Gray [25] (2021)	Case series (n = 9, unverified)	N/R	N/R	CRPS I	N/R
Chokshi [26] (2022)	Case report (n = 1)	49	0/1	CRPS	1 year
Kuah [27] (2022)	Case report (n = 1)	39	0/1	CRPS	4 months
Tombak [28] (2024)	Case report (n = 1)	53	0/1	CRPS I	9 months
Peresa [29] (2024)	Case report (n = 1)	25	0/1	CRPS	2 months

Abbreviations: RSD, reflex sympathetic dystrophy; CRPS, complex regional pain syndrome; M, male; F, female; N/R, not reported.

**Table 2 toxins-17-00350-t002:** Clinical outcomes reported following botulinum toxin injection.

Study (Author, Year)	Outcome(s)	Duration of Effect	Follow-Up Period
Jankovic [5] (1988)	*“Moderate relief of flexion spasm….little improvement in function”*	N/R	N/R
Tarsy [6] (1994)	*“…improved passive flexion at MCP, PIP, and DIP joints, but only mild improvement in voluntary flexion limited to the MP joints of all four digits”*	N/R	N/R
van Hilten [7] (2001)	*“…. botulinum toxin A injections, analgesics, levodopa, trihexyphenidyl, antiepileptics, mannitol infusions, and surgical or chemical sympathectomy proved unrewarding”*	N/R	N/R
Cordivari [8] (2001)	Muscle relaxation, pain relief, posture improvement, functional improvement	6 weeks (1 patient)	N/R
Argoff [9,10] (1999, 2002)	*“…substantial relief of their burning and dysesthetic pain in the affected extremities, as well as normalization of skin color and reduction of any edema that existed before treatment. In addition, the thermal and mechanical allodynia present in all patients before treatment lessened appreciably”*	N/R	4 years
Schrag [11] (2004)	*“Improvement was also reported following botulinum toxin injections in eight patients, but the response ranged from no or transient improvement, to almost complete remission (when combined with positive suggestion)”*	N/R	N/R
Lauretti [12] (2005)	*“…patients presented phalanges and wrist relaxation, reported easy passive physical therapy and pain was classified as 2 (VAS) at passive manipulation.”—“At 8 months evaluation, patients presented 70% and 80% motor and functional recovery of the affected limb.”*	N/R	8 months
Morelet [13] (2005)	*“A motor block followed by botulinic toxin injections was used in one patient, to little effect. In two other patients, the same treatment was followed by a slight decrease in the muscle contraction.”*	N/R	N/R
Safarpour [14] (2010)	Substantial and sustained pain reduction, improved motor function, and enhanced quality of life were reported following BoNT-A treatment, with VAS reduced from 9 to 2–4. Improvements included resolution of finger spasms, reduced allodynia and swelling, and regained ability to perform daily and recreational activities	Up to 3 months	3 years
Kharkar [15] (2011)	Local pain reduction from NRS 8.2 (SD ± 0.8) to 4.5 (SD ± 1.1)	N/R	4 weeks
Mauruc [16]. (2012)	Shoulder abduction: 30°→80°; flexion: 40°→150°; elbow pronation: 60°→90°; supination: −45°→80°; wrist extension: −60°→30°; radial rotation: −20°→0°; pulpo-palmar distances: 80/50/60/70 mm (digits II–V)	N/R	15 days
Vogt [17] (2013)	*“Injection of BoNT/A reduced muscle tone and relieved pain in 5/6 patients, lowering the dosage of analgesic drugs. It was, however, not possible to restore motor function completely.”*	N/R	N/R
Vas [18] (2014)	Unresponsive	N/R	2 years
Fallatah [19] (2014)	“*Repeated follow-up for 3 month showed complete pain relief with full recovery of limb function*.”	N/R	3 months
Drakeley [20] (2016)	Increase in ambulation with axillary crutches from 11 feet to 200 feet	3 months	3 months
Buonocore [21] (2016)	Significant reduction in pain, evoked pain, improved ambulation without crutches	8–10 weeks	30 months
Altonji [22] (2017)	Sufficient for upper limb dystonia, insufficient for lower limb after spread	N/R	N/R
Reddy [23] (2018)	80% reduction in pain, decreased opioid requirement.	11 weeks	3 years
Shenouda [24] (2020)	N/R	N/R	N/R
Gray [25] (2021)	1/20 patients (some with CRPS) with response. No further details	N/R	N/R
Chokshi [26] (2022)	*“The patient had improvement in her pain and ambulation with botulinum toxin injections”*	N/R	N/R
Kuah [27] (2022)	*“After failing physical therapy, botulinum injections, and surgical soft tissue releases, the patient decided to undergo a second surgery.”*	N/R	N/R
Tombak [28] (2024)	VAS reduced from 50 to 30. Improvements in passive and active mobility across elbow, wrist, and MP joints	6 months	6
Peresa [29] (2024)	Improved discomfort from striatal toe, better foot contact in stance phase, reduced stiff-knee gait	N/R	4–6 weeks

Abbreviations: BoNT (A), botulium toxin (-A); VAS, visual analog scale; NRS, numeric rating scale; mm, millimeters; N/R, not reported; MP, metaphalangeal; PIP, proximal interphalangeal; DIP, distal interphalangeal; SD, standard deviation. Note: Italicized quotes are taken directly from cited publications.

## Data Availability

No new data were created or analyzed in this study.

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
