# Peer review of "Intramuscular Botulinum Toxin for Complex Regional Pain Syndrome: A Narrative Review of Published Cases"

_toxins, 2025, doi:10.3390/toxins17070350_

Round 1
Reviewer 1 Report
Comments and Suggestions for Authors
General and detailed comments
The overview of published cases on intramuscular BoNT in CRPS is rather limited, and one may wonder on the validity of the data included and commented. For the reader, the lack of a clear-cut definition of CRPS types is something that is lacking in this manuscript, even if appropriate references are given. In addition, some question can be raised if it is appropriate to include studies in the data analyzed that have severe methodological inconsistencies in the publications reported.
The tables presented have also a number of points that need to be revised. Starting by the authors and publication year, the replacement of authors by the respective reference number and year in parenthesis should make the tables more readable. In the Study design, the indication of a case series of 1, is of no use. Note in the Table 1, column ofAge (Range), the authors can indicate the age and in parenthesis the number of patients. In Table 2, also a number of points should be revised, as indicated here below.
Line 66, SD, standard deviation No indication of the use of SD in this table.
Line 91, “high in” it is not clear the use of “in “
Line 110, The study also provides no specifics regarding, sentence not clear
CRPS type I and II? No definition in the manuscript
Line 220, arthroscopic “knee surger” should be “surgery”
Line 237, “BonT” be consistent use the same abbreviation i.e. BoNT
Lines 257 and 259, define GAS score 0 and GAS score –1 (Goal Attainment Scaling)
Line 270, it should indicate that this Table is in the Appendix section (Appendix, Table 1A)
Line 411, Reference 21 lacks journal publication, page numbering
Line 422, reference 27 needs to be completed, page numbering
Table 2
was classified as 2 (VAS) at passive manipulation.” – “. NOT CLEAR |
“botulinic toxin” TYPO |
VAS scores dropping from 9 to 2–4. |
NRS 8.2 (± 0.8) to 4.5 (± 1.1) SD? |
What is the meaning of italic letters?
botulium toxin(-A);TYPO
In conclusion this manuscript needs more than a simple revision
Comments on the Quality of English LanguageEnglish language is good but some terms are peculiar
Author Response
Esteemed reviewer. Thank you for taking the time to comment and provide feedback on our paper. Your insights and expertise is appreciated. Below we have responded directly to each of your comments.
In addition to edits in response to your comments, we have remedied an oversight in the original manuscript and have added another relevant case report (Tombak, 2024).
Reviewer comment:
The overview of published cases on intramuscular BoNT in CRPS is rather limited, and one may wonder on the validity of the data included and commented. For the reader, the lack of a clear-cut definition of CRPS types is something that is lacking in this manuscript, even if appropriate references are given. In addition, some question can be raised if it is appropriate to include studies in the data analyzed that have severe methodological inconsistencies in the publications reported.
Author response:
We appreciate this thoughtful comment. The aim of our review is precisely to highlight the lack of high-quality, standardized evidence in this area. The inclusion of methodologically heterogeneous and, in some cases, low-quality sources is intentional and serves to illustrate the scattered nature of the existing literature. We believe that synthesizing and critically presenting this body of evidence, however limited, can still offer practical value to clinicians facing complex, treatment-resistant cases of CRPS. The manuscript explicitly comments on methodological inconsistencies to guide cautious interpretation.
Regarding CRPS definitions, we acknowledge that a clear diagnostic framework is important; however, the intent of this review is not to define or rigorously classify CRPS subtypes from published cases. Several of the included publications predate the Budapest criteria or use outdated terminology such as RSD, reflecting historical variation in diagnostic practice—a point we make in the manuscript. For this reason, the inclusion criteria and search terms were necessarily broad. While we agree that not all included patients may meet modern diagnostic criteria, this again underscores the need for improved reporting and standardization in future studies, which the review seeks to call attention to.
We have revised the introduction to better communicate these aims and clarify the rationale for our approach.
Reviewer comment:
The tables presented have also a number of points that need to be revised. Starting by the authors and publication year, the replacement of authors by the respective reference number and year in parenthesis should make the tables more readable. In the Study design, the indication of a case series of 1, is of no use. Note in the Table 1, column ofAge (Range), the authors can indicate the age and in parenthesis the number of patients. In Table 2, also a number of points should be revised, as indicated here below.
Author response: Thank you for your comments. We agree that the legibility of the tables could be improved. To this end, we have implemented the suggested changes.
Reviewer comment:
Line 66, SD, standard deviation No indication of the use of SD in this table.
Author response:
This has been corrected. Thank you for pointing it out.
Reviewer comment:
Line 91, “high in” it is not clear the use of “in “
Author response:
This has been corrected. Thank you once again for pointing it out.
Reviewer comment:
Line 110, The study also provides no specifics regarding, sentence not clear
Author response:
The sentence has been reworded for clarity.
Reviewer comment:
CRPS type I and II? No definition in the manuscript
Author response:
Please see response to first comment. If you feel that a definition is necessary for the reader, we will of course add it.
Reviewer comment:
CRPS type I and II? No definition in the manuscript
Author response:
Please see response to first comment. If you feel that a definition is necessary and would benefit the reader, we will of course add it.
Reviewer comment:
Line 220, arthroscopic “knee surger” should be “surgery”
Line 237, “BonT” be consistent use the same abbreviation i.e. BoNT
Author response:
Misspellings and typos have been corrected. Thank you for your comments.
Reviewer comment:
Lines 257 and 259, define GAS score 0 and GAS score –1 (Goal Attainment Scaling)
Author response:
A definition of the abbreviation GAS has been added.
Reviewer comment:
Line 270, it should indicate that this Table is in the Appendix section (Appendix, Table 1A)
Author response:
This has been corrected.
Reviewer comment:
Line 411, Reference 21 lacks journal publication, page numbering
Line 422, reference 27 needs to be completed, page numbering
Author response:
The references have been expanded to comply with the journal’s standards. Thank you for pointing out this oversight.
Reviewer comment:
Table 2
was classified as 2 (VAS) at passive manipulation.” – “. NOT CLEAR |
“botulinic toxin” TYPO |
VAS scores dropping from 9 to 2–4. |
NRS 8.2 (± 0.8) to 4.5 (± 1.1) SD? |
What is the meaning of italic letters?
botulium toxin(-A);TYPO
Author response:
Italic letters and quotes denote direct quotations from the referenced manuscripts. This is now stated in the table text and hopefully answers the queries on the content of the table, including misspellings and typos. The table text cited in the comment (on VAS) has been reworded for clarity and SD has been added to the parenthesis.
All tables have been updated for legibility and consistency.
We thank the reviewers again for their time and thoughtful comments, which have helped us improve the clarity and rigor of the manuscript.
Reviewer 2 Report
Comments and Suggestions for Authors
Toxins 3694178
This manuscript is just a simple oversight review of the literature on the use of BoNT in Complex Regional Pain Syndrome. No detailed analyses of the publications are presented other than an overview of each report and brief copies of the stated outcomes.
Abstract
“Since the 1980s” – I could find no evidence for the use of BoNT in CRPS earlier than 2001:
Cordivari, C., Misra, V. P., Catania, S., & Lees, A. J. (2001). Treatment of dystonic clenched fist with botulinum toxin. Mov Disord, 16(5), 907–913. https://doi.org/10.1002/mds.1186
The authors have only cited papers from 2018 and 2022 [citations 3 and 4]. This part should therefore be removed.
[Also to note that in lines 31-32 the authors only state three decades of reports, again with no supporting citations]
There are also many other review publications that have considered the use of BoNT for CRPS, going back many years. For example:
Jeynes, L. C., & Gauci, C. A. (2008). Evidence for the use of botulinum toxin in the chronic pain setting--a review of the literature. Pain Pract, 8(4), 269–276. https://doi.org/10.1111/j.1533-2500.2008.00202.x
Safarpour, Y., & Jabbari, B. (2018). Botulinum toxin treatment of pain syndromes -an evidence based review. Toxicon, 147, 120–128. https://doi.org/10.1016/j.toxicon.2018.01.017
Dekhne, A., Goklani, H. D., Doshi, N., Baskara Salian, R., Gandhi, S. K., & Patel, P. (2023). Effectiveness of Botulinum Toxin in the Treatment of Neuropathic Pain: A Literature Review. Cureus, 15(10), e46848. https://doi.org/10.7759/cureus.46848
The manuscript therefore appears to lack the novelty claimed in the Key Contribution and lines 40-41
Overview
To note in the literature selected:
Citations [5,6] makes no mention of BoNT use for pain or CRPS.
Citation [7] only mentions a failed use of BoNT for reflex dystrophy treatment, not CRPS
Citation [10] I could find no mention of Argoff as a contributing author to this 9th World Conference proceedings (based on the published proceedings from the meeting).
Citation [12] seems to make no mention of pain improvements with BoNT
Citation [14] makes no mention of pain relief by BoNT
Citation [19] makes no mention of the effect of a BoNT injection on pain
Citation [21] is incomplete. Line 178 cites [17], which should be [21].
Citation [22] is incorrect. This is an abstract on page e251. The abstract does not mention the effect of BoNT on pain, only on the dystonia.
Citation [26] is unclear about whether the administered BoNT had any effect on pain or whether the effect was only on the dystonic conditions.
Citation 28 makes no mention of the effects of BoNT injections, nor dose or site of injections
Discussion
The authors report in Table 2 what the cited outcomes are, but do not highlight what information on outcomes, especially pain, is missing.
Lines 305-307 Citations needed
Lines 321-322 Citation needed
Conclusions
Line 326 “thorough” should be deleted
Lines 327-329 The authors should state in the Discussion if their findings concur with or disagree with those of the other published reviews of BoNT use in CRPS
I recommend that the layout of the tables be improved as they are not easy to read at present. The citation numbers should be added.
Author Response
Esteemed reviewer. Thank for your taking the time to provide feedback. Your comments are insightful and we have given them due consideration. Below we have provided direct responses to each of your comments.
In addition to edits in response to your comments, we have remedied an oversight in the original manuscript and have added another case report (Tombak, 2024).
Reviewer Comment:
This manuscript is just a simple oversight review of the literature on the use of BoNT in Complex Regional Pain Syndrome. No detailed analyses of the publications are presented other than an overview of each report and brief copies of the stated outcomes.
Author Response:
We thank the reviewer for this comment. The intent of the review is not to conduct a quantitative analysis, but rather to map the available case-based literature and illustrate the lack of high-quality, systematic evidence in this area. By presenting and critically summarizing the published reports, we aim to provide clinicians with a practical overview of the current landscape, however limited, while also underscoring the need for more standardized and methodologically sound studies and case reports. We have revised the introduction to clarify this purpose more explicitly.
Reviewer Comment:
Abstract
“Since the 1980s” – I could find no evidence for the use of BoNT in CRPS earlier than 2001:
Cordivari, C., Misra, V. P., Catania, S., & Lees, A. J. (2001). Treatment of dystonic clenched fist with botulinum toxin. Mov Disord, 16(5), 907–913. https://doi.org/10.1002/mds.1186
The authors have only cited papers from 2018 and 2022 [citations 3 and 4]. This part should therefore be removed.
[Also to note that in lines 31-32 the authors only state three decades of reports, again with no supporting citations]
Author Response:
The reference cited here was published in 2001 and is included and discussed in the paper. The term CRPS was not introduced until 1993 and was not widely adopted until the Budapest meeting in 2004. Previously the syndrome has had many names, including Reflex sympathetic dystrophy, algodystrophy, and causalgia. The papers published before the cited Cordavari paper (with the exception of the abstract by C. Argoff from 1999) all use the term RSD, which at the time was synonymous with what we now call CRPS type I. The introduction has been rewritten to clarify this disparity in terms and the reasons for including papers featuring outdated terminology. We stand by the original wording of the text, including the timespan.
Reviewer Comment:
There are also many other review publications that have considered the use of BoNT for CRPS, going back many years. For example:
Jeynes, L. C., & Gauci, C. A. (2008). Evidence for the use of botulinum toxin in the chronic pain setting--a review of the literature. Pain Pract, 8(4), 269–276. https://doi.org/10.1111/j.1533-2500.2008.00202.x
Safarpour, Y., & Jabbari, B. (2018). Botulinum toxin treatment of pain syndromes -an evidence based review. Toxicon, 147, 120–128. https://doi.org/10.1016/j.toxicon.2018.01.017
Dekhne, A., Goklani, H. D., Doshi, N., Baskara Salian, R., Gandhi, S. K., & Patel, P. (2023). Effectiveness of Botulinum Toxin in the Treatment of Neuropathic Pain: A Literature Review. Cureus, 15(10), e46848. https://doi.org/10.7759/cureus.46848
The manuscript therefore appears to lack the novelty claimed in the Key Contribution and lines 40-41
Author response:
Thank you for your comment. Below we address the cited studies individually:
Jeynes, L. C., & Gauci, C. A. (2008):
While this review does cover CRPS and BoNT-A, it is a) outdated having been published 17 years ago and b) contains a total of 3 lines and 4 references on the subject. The relevant excerpt is reproduced below for clarity.
“7. Complex Regional Pain Syndrome. There are case reports of benefit in patients with Complex Regional Pain Syndrome (CRPS) who were previously unable to open their hands,92,93 as well as reports demonstrating pain relief and functional improvement.94,95”
We believe this brief mention does not substitute for a dedicated, detailed review of the case-based literature, as we provide here.
Safarpour, Y., & Jabbari, B. (2018).
The referenced publication does not include data on CRPS. The condition is mentioned briefly, but in relation to preventative measures after peripheral nerve injury only.
Dekhne, A., Goklani, H. D., Doshi, N., Baskara Salian, R., Gandhi, S. K., & Patel, P. (2023).
CRPS and BoNT-A is mentioned only briefly in the referenced paper, and only cites a single trial on the use of the BoNT-A as an adjuvant for lumbar sympathetic blocks. As such, there is no overlap with our paper.
A recent and potentially overlapping review would be Meta-Analysis of Effectiveness and Safety of Botulinum Toxin in the Treatment of Complex Regional Pain Syndrome by Yu-Chi Su and colleagues (10.3390/life12122037). However, this review pools systematic data from all routes of administration and only includes randomised trials (n=3) and therefore does not address the available case-based evidence, nor is the focus on intramuscular use.
Reviewer Comment:
Citations [5,6] makes no mention of BoNT use for pain or CRPS.
Citation [7] only mentions a failed use of BoNT for reflex dystrophy treatment, not CRPS
Author response:
Please see response to Abstract.
Reviewer Comment:
Citation [10] I could find no mention of Argoff as a contributing author to this 9th World Conference proceedings (based on the published proceedings from the meeting).
Author response:
The referenced abstract can be found on page 50 of the published proceedings, with the number 156. Below are the attached screenshots of the abstract itself, as well the index listing his name for your convenience.
Reviewer comment:
Citation [12] seems to make no mention of pain improvements with BoNT
Citation [14] makes no mention of pain relief by BoNT
Citation [19] makes no mention of the effect of a BoNT injection on pain
Citation [26] is unclear about whether the administered BoNT had any effect on pain or whether the effect was only on the dystonic conditions.
Citation 28 makes no mention of the effects of BoNT injections, nor dose or site of injections
Author response:
Thank you for your feedback. The aim of this paper is not to quantify the effects of BoNT on CRPS, be it pain or functional benefits, but to critically examine the available literature and its lack of consistency. This has been clarified in the introduction section.
Reviewer comment:
Citation [22] is incorrect. This is an abstract on page e251. The abstract does not mention the effect of BoNT on pain, only on the dystonia.
Author response:
While the journal itself lists the proceedings as e122-e335, the abstract itself is, as you correctly point out, published on e251. This has been remedied and the abstract ID has been added.
Reviewer comment:
The authors report in Table 2 what the cited outcomes are, but do not highlight what information on outcomes, especially pain, is missing.
Author response:
To meaningfully highlight what measures are missing from each case report, a consensus on standardized outcome measures must be reached. While some groups, namely the COMPACT initiative, are actively trying to homogenize trial outcomes, to our knowledge no such recommendations exist for case reports. In the discussion, we put forth some recommendations as to what should be included in future publications. However, we do not feel that the addition of an exhaustive list of relevant details not included in each study would contribute to Table 2.
Reviewer comment:
Lines 305-307 Citations needed
Lines 321-322 Citation needed
Author response:
Relevant citations have been added. Thank you for your comment.
Reviewer comment:
Conclusions
Line 326 “thorough” should be deleted
Author response:
Thank you for your comment. This has been corrected.
Reviewer comment:
Lines 327-329 The authors should state in the Discussion if their findings concur with or disagree with those of the other published reviews of BoNT use in CRPS
Author response:
Thank you for your response. We have added a brief section describing previously published reviews and their findings in the Discussion.
Reviewer comment:
I recommend that the layout of the tables be improved as they are not easy to read at present. The citation numbers should be added.
Author response:
All tables, including those in the appendix, have been updated for better legibility. Reference numbers have been added.
We thank the reviewers again for their time and thoughtful comments, which have helped us improve the clarity and rigor of the manuscript.
Round 2
Reviewer 1 Report
Comments and Suggestions for Authors
Authors have modified in parts the manuscript, and there are still a few points that need further revision. These are the following:
- Line 428, Abbreviations should be maintained only when used 5 or more times, and should be defined the first time they appear in text or Tables.
In the article most abbreviations are used in Tables, and should be defined there. Is it necessary to have a page of Abbreviations?
-
- Is Table A2, necessary? The information of this table should be given just in two lines of text.
- References should be presented in a homogeneous manner. Sometimes are used with the abbreviated name of journal, other times with the complete name of the journal. Please be consistent.
- Line 144 should be "A frequently"
- In Table 2, at the level of Morelet [13] (2005) replace “botulinic” by botulinum
- Line 402, insert a space between vascular and [40]
- In reference 15, no DOI, line 483, Therefore, add PMID: 21587336
- Line 494, Reference 20 The journal in which was published and the DOI should be indicated
Author Response
Reviewer comment:
Line 428, Abbreviations should be maintained only when used 5 or more times, and should be defined the first time they appear in text or Tables.
Author response:
The separate section containing all abbreviations used is per the journal’s author template. While we agree with you in principle, we try to conform to the standard of the journal as closely as possible. We leave it up to the editor whether this inclusion is necessary. Thank you for your comment.
Reviewer comment:
2. Is Table A2, necessary? The information of this table should be given just in two lines of text
Author Response:
We agree. Table A2 has been removed and a short text added to the relevant section under Discussion.
Reviewer comment:
References should be presented in a homogeneous manner. Sometimes are used with the abbreviated name of journal, other times with the complete name of the journal. Please be consistent.
Author response:
This has been remedied and the references updated.
Reviewer comment:
Line 144 should be "A frequently"
Author response:
This has been corrected.
Reviewer comment:
In Table 2, at the level of Morelet [13] (2005) replace “botulinic” by botulinum
Author response:
This is a direct quote from the publication and the use of “botulinic” is the author’s, not ours. Changing the spelling would therefore misquote the article.
Reviewer comments:
- Line 402, insert a space between vascular and [40]
- In reference 15, no DOI, line 483, Therefore, add PMID: 21587336
- Line 494, Reference 20 The journal in which was published and the DOI should be indicated
Author response:
These oversights have all been corrected. Thank you for your time and attention to detail.
Reviewer 2 Report
Comments and Suggestions for Authors
The revised version has addressed all the issues that I raised during my first review
Author Response
Reviewer comment:
The revised version has addressed all the issues that I raised during my first review
Author response:
We thank you for your time and commitment and look forward to working with you again in the future.